

# Effect of soil coarseness on soil base cations and available micronutrients in a semi-arid sandy grassland

L. Lü[1,2], R. Wang[1,*], H. Liu[1,3], J. Yin[1,4], Z. Wang[1], Y. Zhao[2], G. Yu[2], X. Han[1], Y. Jiang[1]

[1] State Key Laboratory of Forest and Soil Ecology, Institute of Applied Ecology, Chinese

Academy of Sciences, Shenyang 110016, China

[2] Institute of Sand Fixation and Utilization, Liaoning Academy of Agricultural Sciences,

Fuxin 123000, China

[3] Key Laboratory of Regional Environment and Eco-remediation, College of Environment,

Shenyang University, Shenyang 110044, China

[4] University of Chinese Academy of Sciences, Beijing 10049, China

*Correspondence to:* R. Wang (ruzhenwang@iae.ac.cn)

## Abstract

Soil coarseness is the main process decreasing soil organic matter and threatening the

productivity of sandy grasslands. Previous studies demonstrated negative effect of soil

coarseness on soil carbon storage, but less is known about how soil base cations

(exchangeable Ca, Mg, K, and Na) and available micronutrients (available Fe, Mn, Cu,

and Zn) response to soil coarseness. In a semi-arid grassland of northern China, a field

experiment was initiated in 2011 to mimic the effect of soil coarseness on soil base

cations and available micronutrients by mixing soil with different mass proportions of

sand: 0% coarse elements (C0), 10% (C10), 30% (C30), 50% (C50), and 70% (C70).

Soil coarseness significantly increased soil pH in three soil depths of 0-10 cm, 10-20

cm and 20-40 cm with the highest pH values detected in C50 and C70 treatments. Soil

fine particles (smaller than 0.25 mm) significantly decreased with the degree of soil

coarseness. Exchangeable Ca and Mg concentrations significantly decreased with soil

coarseness degree by up to 29.8% (in C70) and 47.5% (in C70), respectively, across

three soil depths. Soil available Fe, Mn and Cu significantly decreased with soil

coarseness degree by 62.5%, 45.4% and 44.4%, respectively. As affected by soil



coarseness, the increase of soil pH, decrease of soil fine particles (including clay), and decline in soil organic matter were the main driving factors for the decrease of exchangeable base cations (except K) and available micronutrients (except Zn) through soil profile. Developed under soil coarseness, the loss and redistribution of

base cations and available micronutrients along soil depths might pose threat to ecosystem productivity of this sandy grassland.

## 1   Introduction

Dryland ecosystems, accounting for 41% of the total land area of the world are prone

to desertification which would result in soil coarseness (Cerdàet al., 2014; Wang et al., 2015a). Dryland ecosystems represent 25% of land surface area in Latin America with 75% of them having desertification problems (Torres et al., 2015). Desertified land area has been reported to reach 45.6 million km$^2$ (Torres et al., 2015) and accounted for 74% of total dryland area (61.5 million km$^2$) with more than 100 countries and

$8.5 \times 10^8$ people being affected (Miao et al., 2015; Vieira et al., 2015; Wang et al., 2015a). Desertification exerts large impact on social and economic resources (Beyene, 2015; Escadafal et al., 2015). Areas susceptible to desertification tend to be marked by socioeconomic inequality and have low human development index in Brazil (Vieira et al., 2015). Desertification was reported to cause economic losses of up to €6 billion in

Northern China in the year of 2005 (Miao et al., 2015).

     In China, most of the grasslands have undergone degradation and desertification with 50% distributed in the agro-pastoral transition zone of northern China (Wang et al., 2015a; Yan and Cai, 2015). Desertification and wind erosion processes induced by over-exploitation of land resources in fragile areas and overgrazing in arid and

semi-arid rangelands have contributed to increased soil coarseness (Yan and Cai, 2015). Together with reduction of plant cover, increased soil coarseness contribute to loss of agricultural productivity, environmental deterioration, and associated social and economic disruptions (Vieira et al., 2015; Xie et al., 2015). In this case, it is urgent to combat desertification and study the causes, processes, consequences, and

mechanisms of soil coarseness (Xu et al., 2012; Weinzierl et al., 2015).



Soil base cations are not only essential nutrient cations for both plants and soil microbes, but also serve as one of the main mechanisms of soil acid buffering capacity (Lu et al., 2014) as well as a good indicator of soil fertility (Zhang et al., 2013). Micronutrient availabilities essentially affect terrestrial net primary production, plant quality, and consequently food and forage supply worldwide (Cheng et al., 2010; Marques et al., 2015). Current research about desertification and soil coarseness mainly focus on its effects on degradation of forest and grasslands due to logging and overgrazing (Conte et al., 1999; Cao et al., 2008), C and N depletion in soils and plant components (Zhou et al., 2008; Bisaro et al., 2014), soil compaction and erosion risk (Allington and Valone, 2010), and soil physical properties of particle size distributions (Su et al., 2004; Huang et al., 2007). However, less is known about the changes in soil base cations and availabilities of micronutrient during dryland desertification and soil coarseness.

Soil coarseness is suggested to cause decrease of soil silt and clay contents (Zhou et al., 2008), decline in soil C and nutrient (such as N and P) concentrations (Xie et al., 2015), and losses in species diversity and productivity (Zhao et al., 2006; Huang et al., 2007). As biogeochemical cyclings of base cations and micronutrients are largely controlled by soil organic matter (SOM) (complexation and chelation) (Sharma et al., 2004) and properties of soil mineral (reversible sorption and desorption processes) (Jobbágy et al., 2004), decrease of SOM and soil fine particles would potentially decrease soil base cations and micronutrient availability. Also, the changes in SOM along soil depth could shape the vertical distribution of base cations and available micronutrients (Sharma et al., 2004).

The Horqin Sandy Land, or Horqin Sandy Grassland is an important part of Inner Mongolia grassland and one of the main sandy areas in northern China covering approximately 43,000 km$^2$ (Li et al., 2004). In Horqin region, the soils are prone to aeolian soil erosion and soil coarseness especially when natural sandy grassland is converted into farmland (Li et al., 2004). To examine the effect of soil coarseness during desertification on the concentrations of soil base cation (exchangeable Ca, Mg, K, and Na) and available micronutrient (Fe, Mn, Cu, and Zn) of this region, we set up




a field experiment in Zhanggutai by mixing the soil with different mass proportions of sand: 10% (light soil coarseness), 30% (moderate soil coarseness), 50% (heavy soil coarseness), and 70% (severe soil coarseness). We hypothesized that both soil base cations and available micronutrients would decrease with the increasing degree of soil

coarseness due to the decrease of SOM and soil fine particles. We also expect that soil base cations and available micronutrients would decrease with soil depth.

## 2 Materials and methods

### 2.1 Study area

The study was conducted at the Desertified Grassland Restoration Research Station maintained by Institute of Sand Fixation and Utilization, Liaoning Academy of Agricultural Sciences. The study site (42 °43′N and 122 °22′E, elevation 226.5 m a.s.l.) was located in the southeast of Horqin Sandy Land, near Zhanggutai Town, Zhangwu County, Liaoning Province, China. Productive grasslands from Zhanggutai County

have undergone severe desertification due to intense cultivation, overgrazing and increased population (Li et al., 2000; Chen et al., 2005). The mean annual temperature is 6.2 °C and mean annual precipitation is about 450 mm being a semi-arid region (Chen et al., 2005). Soil texture of the experiment site is sandy soil with 99.32 ±0.13 % sand, 0.45 ±0.14 % silt, and 0.23 ±0.02 % clay (means ±standard deviation, data

measured from control soil). The soil type is classified as a Aeolic Eutric Arenosol according to the FAO classification (IUSS Working Group WRB, 2014). This area constitutes an agro-pastoral ecotone which is severely degraded due to excessive cultivation and grazing (Chen et al., 2005).

### 2.2 Experimental design

In May 2011, a complete randomized design was applied to the site. Thirty 4 m × 4 m plots were established for five treatments with six replicates per treatment. Adjacent plots were separated by 1 m buffer zone and PVC plates to prevent water and nutrient exchanges. A certain mass proportion of 2 mm- sieved river sand (siliceous, pH 7.5 ±

0.2) was mixed with native soil for each of three depths (0-20 cm, 20-40 cm, and



40-60 cm). Three soil depths were considered in this study to study effect of soil coarseness on soil properties of plant root layer (0-20 cm), transition layer of plant roots (20-40 cm), and transition layer of soil genesis (40-60 cm). To mix the river sand and native soils evenly in each plot, soils of each depth were digged out and

mixed with the sand by agitators in the same mass proportion separately. The soils were refilled back to the field in respective depths after mixture. The proportions are 0, 10%, 30%, 50%, and 70% to mimic different soil coarseness degrees or intensities: control grassland without soil coarseness (C0), light soil coarseness (C10), moderate soil coarseness (C30), heavy soil coarseness (C50), and severe soil coarseness (C70),

respectively. In August 2012, 0-5 cm soils of all plots were taken out and autoclaved at 105 ℃ for 3 h to deactivate the seeds and then refilled back. This was adequate to prevent the reproduction of original plants. In July 2013, plant community was transplanted from local natural grassland according to its species composition by point quadrats (Goodall, 1952). Plant community composition was investigated in a

permanent quadrat of 1 m × 1 m at August of 2014 and 2015 (unpublished data). The chemical characteristics of the 0-10 cm soil are given in Table 1.

### 2.3 Soil sampling and chemical analysis

In October 2015 (i.e. after 2 years of plant community settled), a composite soil

sample was taken from three randomly selected locations within each plot from three soil layers of 0-10 cm, 10-20 cm, and 20-40 cm, respectively. Fresh soil samples were sieved through 2 mm screen and visible plant roots were taken out. After transportation to laboratory, the soils were air-dried and a subsample of the soil was ground for C and N analysis.

### 2.3.1 Soil pH and particle size distribution

Soil pH was determined in a 1:2.5 (w/v) soil-to-water extract of soil samples from all treatments with a PHS-3G digital pH meter (Precision and Scientific Corp., Shanghai, China). Soil particle size distribution was determined by the pipette method in a

sedimentation cylinder, using Na-hexamethaphosphate as the dispersing agent (Zhao





et al., 2006). Proportion of soil fine particles (<0.25mm) were calculated by summing up the proportions of fine sand, silt and clay in this study.

### 2.3.2 Soil base cations (Ca, Mg, K, Na) and available micronutrients (Fe, Mn, Cu, Zn)

Soil base cations were determined using the $CH_3COONH_4$-extraction method according to Ochoa-Hueso et al. (2014). Briefly, 2.5 g of soil sample was extracted with 1 M $CH_3COONH_4$ (pH 7.0) with a soil:extractant ratio of 1:20 (w/v) and shaken at 150 rpm for 30 min. After filtration with Whatman no. 2V filter paper, the concentrations of soil base cations were determined by atomic absorption spectrometer (AAS, Shimazu, Japan).

Available Fe, Mn, Cu and Zn were extracted by diethylenetriaminepentaacetic acid (DTPA) according to method of Lindsay and Norvell (1978). Briefly, 10 g of soil samples was mixed with 20 ml 0.005 M DTPA + 0.01 M $CaCl_2$ + 0.1 M triethanolamine (TEA) (pH 7.0). The slurry was shaken at 180 rpm for 2 h and filtered through Whatman no. 2V filter paper. The concentrations of available micronutrients were analyzed by AAS.

### 2.4 Statistical analyses

The normality of data was tested using the Kolmogorov-Smirnov test, and homogeneity of variances using Leven's test. Effects of soil coarseness on soil pH, fine particles, base cations and available micronutrients were determined by one-way ANOVA. Multiple comparisons with Duncan design were performed to determined difference in soil parameters among soil coarseness degrees. Pearson correlation analysis was used to examine the relationship among soil parameters. Multivariate linear regression analyses (stepwise removal) were conducted to determine variables that made significant contributions to variance of soil base cations and available micronutrients. All statistical analyses were performed in SPSS 16.0 (SPSS, Inc., Chicago, IL, U.S.A) and statistical significance was accepted at $P < 0.05$.





## 3 Results

### 3.1 Soil pH

Soil coarseness significantly increased soil pH by up to 8.8% across three soil depths (Fig. 1a; Table 2). For both 0-10 cm and 10-20 cm soils, the highest soil pH was

detected in C70 (7.3 and 7.4, respectively) and C50 (7.2 and 7.3, respectively) soils, which were followed by C30 and C10 soils (Fig 1a). Significant and positive overall effect of soil depth was detected on soil pH (Fig. 1a; Table 2). For both C0 and C50 treatments, soil pH of 10-20 cm and 20-40 cm was significantly higher as compared to that of 0-10 cm soil (Fig. 1a). Soil pH in10-20 cm of C10 and C30 was significantly

higher than that in 0-10 cm of C10 and C30, respectively (Fig. 1a). Significant interactive effect of soil coarseness and soil depth was found on soil pH (Table 2).

Proportions of soil fine particles (< 0.25 mm) were determined in 0-10 cm soil. Soil fine particles significantly decreased with soil coarseness degree by 6.3% (for treatment of C10), 17.7% (C30), 34.1% (C50), and 55.6% (C70) as compared to C0

(Fig. 1b). The lowest proportion of soil fine particles was detected in C70 (39.1%), and followed by C50 (58.1%) (Fig. 1b). Proportion of clay particles significantly decreased under soil coarseness (Fig. S1).

### 3.2 Soil base cations

Across three soil depths, soil coarseness significantly decreased both exchangeable Ca and Mg concentrations by up to 29.8% and 47.5%, respectively, as compared to C0 (Fig. 2a,b). Both exchangeable Ca and Mg concentrations were the lowest in the C70 and followed by C50 as compared to C0 in all soil depths (Fig. 2a,b). Soil depth significantly decreased soil exchangeable Mg, while showed no effect on

exchangeable Ca (Table 2). Both soil coarseness and soil depth had no impact on soil exchangeable K (Fig. 2c). At 0-10 cm, C50 and C70 significantly decreased soil exchangeable Na by 22.3% and 24.2%, respectively, as compared to C0 (Fig. 2d). Soil exchangeable Na did not change with soil depth (Fig. 2d, Table 2).

**3.3 Soil available micronutrients**



Soil available Fe significantly decreased with soil coarseness degree by as much as 17.1% in C10, 22.0% in C30, 36.6% in C50 and 62.5% in C70 across three soil depths (Fig. 3a). Soil coarseness significantly decreased soil available Mn for 0-10 cm (by up to 17.3% in C70) and 10-20 cm (by up to 45.4% in C70) soils (Fig. 3b). Both soil

available Fe and Mn significantly decreased with soil depth (Fig. 3a,b; Table 2). Significant negative desertification effect was detected on soil available Cu by 14.7% - 44.4% as compared to C0 in 0-10 cm soil (Fig. 3c). For both C30 and C50 treatments, soil available Cu concentration of 10-20 cm soil was significantly higher than that in 0-10 cm and 20-40 cm soils. Soil available Zn concentration was not

affected by soil coarseness but it decreased with soil depth (Fig. 3d; Table 2).

### 3.4 Regression analyses between soil parameters

All regression analysis were conducted for 0-10 cm soil as the data of fine particles (< 0.25 mm) were only available for 0-10 cm soil. At 0-10 cm soil, soil pH significantly

and negatively correlated with exchangeable Ca, Mg and Na, and with available Fe, Mn and Cu (Table 3). Soil fine particles (< 0.25 mm) significantly and positively correlated with exchangeable Ca, exchangeable Mg, exchangeable Na, available Fe, available Mn, and available Cu (Table 3). The SOC significantly and positively correlated with exchangeable Ca, Mg, Na, available Fe, Mn, and Cu (Table 3).

According to multiple regression models, change of soil fine particles explained 65.5%, 75.7%, 31.4%, 24.0% of variations in exchangeable Ca, Mg, Na, and available Mn (Table 3). Soil pH explained 75.7% of variation in available Fe (Table 2). The SOC explained 59.3% of variation in available Cu (Table 2).

### 4 Discussion

#### 4.1 Effect of soil coarseness on soil base cations and available micronutrients

Significant decrease in exchangeable Ca and Mg concentrations in three soil depths and exchangeable Na in 0-10 cm soil as affected by soil coarseness partially supported our first hypothesis. The decrease of exchangeable Ca, Mg and Na might be due to

increase of soil pH under soil coarseness as suggested by the significant and negative





correlation between soil pH and exchangeable Ca, Mg and Na (Table 3). Indeed, with the increase of soil pH, soil base cations (such as $Ca^{2+}$ and $Mg^{2+}$) and available micronutrients ($Fe^{2+}$, $Mn^{2+}$ and $Cu^{2+}$) would precipitate with $OH^-$ (McLean, 1982) resulting in the decrease of soil base cations and available micronutrients under soil

coarseness.

Soil fine particles (< 0.25 mm), especially clay inside these fine particles were suggested to provide additional binding surfaces for exchangeable base cations and available micronutrients (Beldin et al., 2007). Confirmed by the negative correlation of soil fine particles with both base cations (exchangeable Ca, Mg and Na) available

micronutrients (Fe, Mn and Cu), the decrease of soil fine particles and clay content (Fig. S1) might also contribute to lower base cations and available micronutrients under soil coarseness. Consistent with our findings, previous studies also suggested that the decrease of soil fine particles and increase of soil coarseness resulted in loss of SOM as well as reduction in the nutrient storage (Lopez, 1998; Zhao et al., 2006;

Zhou et al., 2008).

As the essential role of SOM in retaining base cations and micronutrients by its functional groups (Oorts et al., 2003), significantly lower soil base cations and micronutrients would possibly due to lower C (the largest component of SOM) concentration in coarsen soils. This can be further enhanced by the significant positive

correlation of soil C with both base cations and micronutrients (Table 3). Consistently, Vittori Antisari et al. (2013) reported that humified organic compounds in soil could retain base cations and decrease their leaching from soils. As compared to higher soil coarseness degree, higher soil microbial activities (unpublished data) under conditions of lower soil coarseness degree could promote humification or microbial-processing

of the SOM (Wang et al., 2015b), potentially increasing the availability of functional groups to complex with the base cations and micronutrients. Due to the fact of reduction in ecosystem productivity under cation deficiencies (Lawrence et al., 1995; Cheng et al., 2010), the loss of base cations and available micronutrients as developed under soil coarseness could constrain both plant growth and pasture productivity of

this nutrient poor sandy ecosystem.



### 4.2 Effect of soil depth on base cations and available micronutrients

The hypothesized decrease of exchangeable base cations and available micronutrients with soil depth was partially supported as only exchangeable Mg (Fig. 1b), available Fe (Fig. 2a), Mn (Fig 2b) and Zn (Fig. 2d) decreased with soil depth. Vertical distribution of soil nutrients can be influenced by two opposite processes, leaching and biological cycling (such as plant absorption) (Truggill, 1988). Being a ubiquitous process in ecosystems, plant absorption of nutrients can transport soil elements aboveground and return the litterfall to soil surface (Stark, 1994). Especially in this sandy land or desertified grassland, plants tend to accumulate SOM or nutrients to form 'island of fertility' (Cao et al., 2008). In these sandy soils, leaching is also an essential process in shaping the vertical distribution of soil nutrients (Truggill, 1988). As leaching moves nutrients downward while biological cycling moves them upward (Jobbágy and Jackson, 2001), the unchanged Ca, K and Na concentrations might be the combining effects of leaching and biological cycling. Our results are in contrast with previous studies suggesting that ecosystem were more capable to retain K than other base cations (Nowak et al., 1991; Jobbágy and Jackson, 2001). In this case, it is obvious that many environmental factors, like soil types and plant community composition can be drivers for the vertical distribution of base cations and micronutrients. Stronger effect of plant absorption than leaching might contribute to the shallower distribution of exchangeable Mg (Fig. 1b), available Fe (Fig. 2a), Mn (Fig. 2b) and Zn (Fig. 2d). The dominant role of plant cycling in determining the vertical distribution of Mg, Fe, Mn and Zn might illustrate that these elements were scarcer and more limiting nutrients for plant growth in this semi-arid sandy ecosystem (Jobbágy and Jackson, 2001).

### 5 Conclusions

The results showed that grassland soil coarseness decreased soil base cations of exchangeable Ca, Mg and Na as well as available micronutrients of Fe, Mn and Cu. The loss of SOM, decrease of soil fine particles, and increase of soil pH were the



main driving factors for the decrease of base cations and micronutrient availability as affected by soil coarseness. Unchanged concentrations of exchangeable Ca, K and Na along the soil depth might result from the balance between plant cycling and leaching effects. The dominant role of plant cycling over leaching shaped the shallower

distribution of exchangeable Mg as well as available Fe, Mn and Zn. The reduction and re-distribution of soil base cations and available micronutrients would potentially influence soil fertility and plant productivity in this desertified grassland ecosystem.

**Author contribution**

Z. Wang, G. Yu, and X. Han designed the experiments; and L. Lü and Y. Zhao carried them out. H. Liu and J. Yin help to do the laboratory analysis. L. Lü and R. Wang prepared the manuscript with contributions from all authors. Y. Jiang helped to revise the manuscript.

**Acknowledgments**

This work was financially supported by the National Natural Science Foundation of China (41371251).

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



**Table 1** Mean and range of soil chemical characteristics for 0-10 cm soil in different soil coarseness degrees from 0% sand addition (C0) to 70% (C70).

|                                              | Range of mean from C0 to C70 |
| -------------------------------------------- | ---------------------------- |
| Soil organic carbon (g kg soil$^{-1}$)       | 4.1-2.7                      |
| Total nitrogen (g kg soil$^{-1}$)            | 0.48-0.22                    |
| Dissolved organic carbon (mg kg soil$^{-1}$) | 66.3-53.9                    |
| Microbial biomass C (mg kg soil$^{-1}$)      | 104.3-60.4                   |
| Electric conductivity (μs cm$^{-1}$)         | 54.7-44.7                    |





**Table 2** Results ($F$ values) of two-way ANOVAs on the effect of soil depth (D),
treatments of soil coarseness degrees (T), and their interactions on soil base cations
(exchangeable Ca, Mg, K, and Na) and available micronutrients (Fe, Mn, Cu, and Zn).

|  | pH | Ca | Mg | K | Na | Fe | Mn | Cu | Zn |
|---|---|---|---|---|---|---|---|---|---|
| D | 17.33$^{**}$ | 0.99 | 4.54$^{*}$ | 0.74 | 0.22 | 135.67$^{**}$ | 99.63$^{**}$ | 9.69$^{**}$ | 22.13$^{**}$ |
| T | 31.74$^{**}$ | 50.26$^{**}$ | 57.22$^{**}$ | 1.61 | 2.35 | 41.51$^{**}$ | 12.11$^{**}$ | 10.60$^{**}$ | 0.62 |
| D×T | 4.12$^{**}$ | 1.49 | 3.81$^{**}$ | 0.64 | 5.02$^{**}$ | 2.97$^{**}$ | 2.49$^{*}$ | 3.40$^{**}$ | 0.83 |

$^{*}$ Significance level at $P < 0.05$.

$^{**}$ Significance level at $P < 0.01$.



**Table 3** Regression statistics relating soil base cations (exchangeable Ca, Mg, K, and Na) and available micronutrients (Fe, Mn, Cu, and Zn) to soil pH, soil fine particles (<0.25 mm) and soil organic carbon (SOC).

|    | Soil pH | < 0.25mm | SOC | Multiple |
|----|---------|----------|-----|----------|
| Ca | -0.67 | 0.81[**] | 0.73 | 0.81 |
| Mg | -0.75 | 0.87[**] | 0.72 | 0.87 |
| K | — | — | — | — |
| Na | -0.54 | 0.56[**] | 0.56 | 0.56 |
| Fe | -0.87[**] | 0.80 | 0.74 | 0.87 |
| Mn | -0.42 | 0.49[**] | 0.45 | 0.49 |
| Cu | -0.68 | 0.72 | 0.77[**] | 0.77 |
| Zn | — | — | — | — |

Values are $R$ statistics for significant ($P < 0.05$) linear regressions. Multiple is $R$ values for multiple regressions (stepwise removal) of soil base cation and micronutrients with soil pH, <0.25 mm fine particles, and SOC. [**] indicates variables that make significant contributions to the multiple linear regressions.





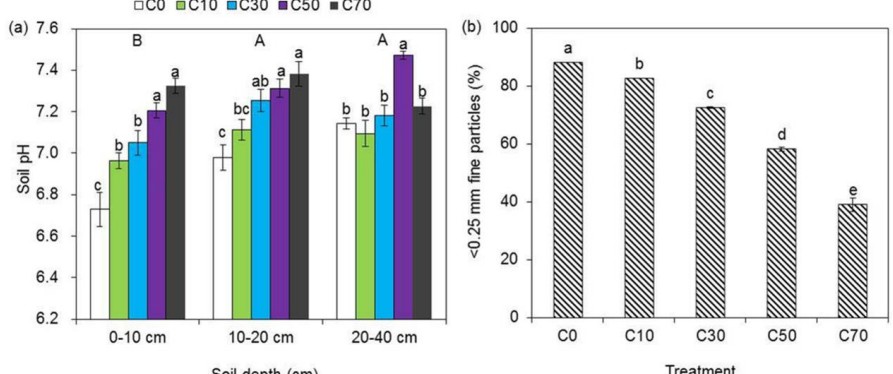

**Fig. 1** Soil pH values for three soil depths (a) and proportion of soil fine particles (<

5    0.25 mm) for 0-10 cm soil in different soil coarseness degrees of 0% sand addition

(C0), 10% (C10), 30% (C30), 50% (C50) and 70% (C70). Data represent mean $\pm$ SE

(n=6). Letters indicate significant differences among treatments (lowercase letters)

and differences among soil depths when averaging across all treatments (capital

letters).



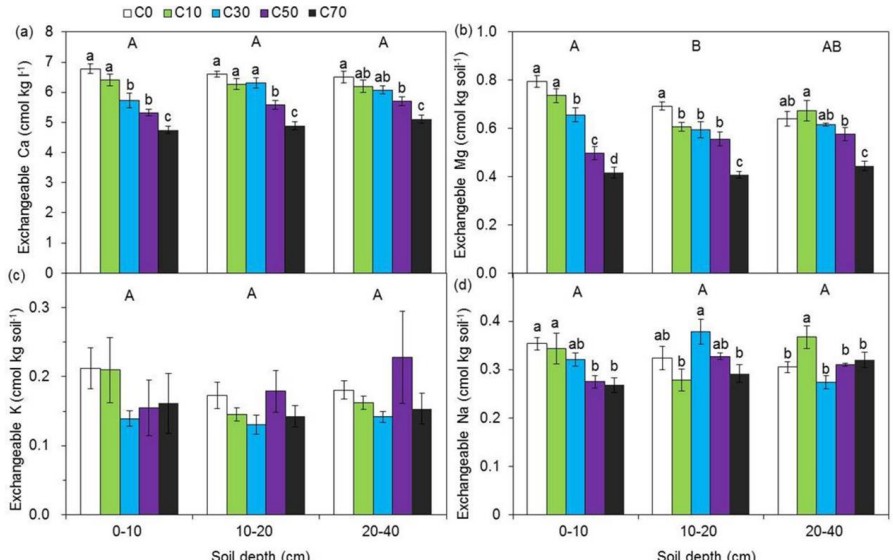

**Fig. 2** Soil base cations of exchangeable Ca (a), Mg (b), K (c) and Na (d) for three soil depths in different soil coarseness degrees of 0% sand addition (C0), 10% (C10), 30% (C30), 50% (C50) and 70% (C70). Data represent mean ±SE (n=6). Letters indicate significant differences among treatments (lowercase letters) and differences among soil depths when averaging across all treatments (capital letters).



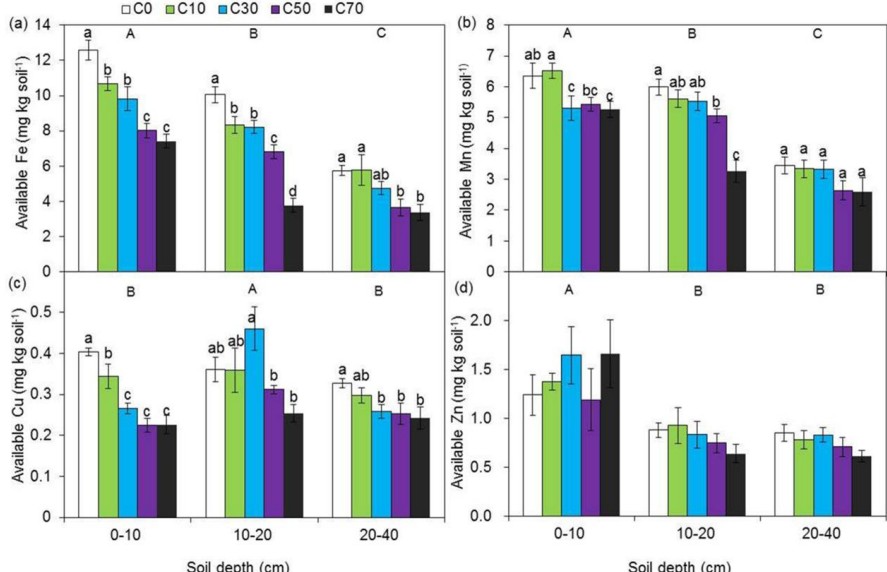

**Fig. 3** Soil available micronutrients of available Fe(a), Mn (b), Cu (c) and Zn (d) for

5    three soil depths in different soil coarseness degrees of 0% sand addition (C0), 10%

(C10), 30% (C30), 50% (C50) and 70% (C70). Data represent mean ± SE (n=6).

Letters indicate significant differences among treatments (lowercase letters) and

differences among soil depths when averaging across all treatments (capital letters).

