# Peer review of "Effect of soil coarseness on soil base cations and available"

_Solid Earth, 2016_

## Referee Comment (RC1) · Anonymous Referee #1 · 29 Jan 2016

This manuscript presents an interesting experiment conducted in northern China examining the effect of soil coarsening on soil base cations and available micronutrients. The paper is well-written, clearly organized and easy to follow by the reader. The topics fit within the scope of the journal, although the the authors should highlight how the geochemical data influences soil formation.

I have some comments on several sections:

- In the Introduction the processes inducing soil coarsening around the world are briefly mentioned, but then they are not mentioned along the text. It would be important to discuss which factors are driving soil coarsening in the study area. This would help the reader to better understand the regional setting and, consecuently, the findings presented in this research. - In the Study area the authors should include a reference

to some other factors which affect desertification in this region, such as vegetation, lithology (bedrock, sediments?) or aeolian processes (significant?). - Are different soil layers along the sections? If it is homogeneous, mention it. If soil ayer show distinct features, this might have affeted your sampling and therefore your experiment. Please clarify it. - Another thing which you may have influenced your data at depth are cryogenic processes. The area records freeze-thaw cycles during at least 4-5 months per year. Freeze-thaw cycles affect the vertical structure of the soil through cryoturbation activity. How this process may have affect your data? - You present nice data about soil base cations and available micronutrients at different depths, but you do not discuss how they influence soil formation processes (pedogenesis).

Tables are OK, but the paper would substantially improve with 1/2 new figures inluding the site location, soil sections, etc.

Line 41 add comma after world Line 97 change expect to "expected" L. 109 after mm, change to "which defines the area as semi-arid"
* * *

---

## Author Comment (AC1) · 3 Feb 2016

Comments from Reviewer #1: Comment: In the Introduction the processes including soil coarsening around the world are briefly mentioned, but they are not mentioned along the text. It would be important to discuss which factors are driving soil coarsening in the study area.

Response: We added information related to soil coarseness in the context. Please see Line 41-42, Line 55-64, Line 70-77, Line 78-81, and Line 88-92. And now, the driving factors of soil coarseness in this area can be found in Line 55-59 and Line 90-92.

Comment: In the Study area the authors should induce a reference to some other factors which affect desertification in this region, such as vegetation, lithology (bedrock, sediments?) or aeolian processes (significant?)
[Figure]

Response: The information with reference to other factors has been added. Please find in Line 109-114.

Comment: Are soil layers different along the sections? If it is homogenous, mention it. If soil layer show distinct features, this might have affected your sampling and therefore your experiment. Please clarify it.

Response: We agree with the suggestion. This information has been added in Line 126.

Comment: Another thing which you may have influenced your data at depth are cryogenic processes. The area records freeze-thaw cycles during at least 4-5 months per year. Freeze-thaw cycles affect the vertical structure of the soil through cryoturbation activity. How this process may have affected your data?

Response: The reviewer brings up a good point and we have expanded our speculation related to freeze-thaw cycles. Please see Line 296-299.

Comment: You present nice data about soil base cations and available micronutrients at different depths, but you do not discuss how they influence soil formation processes (pedogenesis).

Response: This information has been added in Line 304-307.

Comment: Tables are OK, but the paper would substantially improve with 1/2 new figures including the site location, soil sections, etc.

Response: This figure has been added as Fig. 1. Mr. Xiao helps to create this figure, so we added him as a co-author.

Comment: Line 41 add comma after world; Line 97 change expect to 'expected'; Line 109 after mm, change to 'which defines the area as semi-arid'.

Response: These has been corrected in the context. Please see Line 41, Line 100, and Line 115.
[Figure]

Please also note the supplement to this comment:
http://www.solid-earth-discuss.net/se-2016-18/se-2016-18-AC1-supplement.pdf
[Figure]

N

**Zhanggutai County**

**Liaoning Province**

50 100    200
KM

**Fig. 1.** Fig. 1 Location of the experimental site.

[Figure]

[Figure]

**Fig. 2.** Fig. 2 Soil pH values for three soil depths (a) and proportion of soil fine particles (< 0.25 mm) for 0-10 cm soil in different soil coarseness degrees of 0% sand addition (C0), 10% (C10), 30% (C30),

[Figure]

**Fig. 3.** Fig. 3 Soil base cations of exchangeable Ca (a), Mg (b), K (c) and Na (d) for three soil depths in different soil coarseness degrees of 0% sand addition (C0), 10% (C10), 30% (C30), 50% (C50) and 70% (

[Figure]

**Fig. 4.** Fig. 4 Soil available micronutrients of available Fe(a), Mn (b), Cu (c) and Zn (d) for three soil depths in different soil coarseness degrees of 0% sand addition (C0), 10% (C10), 30% (C30), 50% (C50)

[Figure]

**Fig. 5.** Fig. S1 Proportion of soil clay particles for 0-10 cm soil in different soil coarseness degrees of 0% sand addition (C0), 10% (C10), 30% (C30), 50% (C50) and 70% (C70). Data represent mean ± SE (n=6).

**Supplement:**

**Effect of soil coarseness on soil base cations and available**

**micronutrients in a semi-arid sandy grassland**

Linyou Lü[1,2], Ruzhen Wang[1,*], Heyong Liu[1,3], Jinfei Yin[1,4], Jiangtao Xiao[1,4],

Zhengwen Wang[1], Yan Zhao[2], Guoqing Yu[2], Xingguo Han[1], Yong Jiang[1]

[1] State Key Laboratory of Forest and Soil Ecology, Institute of Applied Ecology,

Chinese Academy of Sciences, Shenyang 110016, China

[2] Institute of Sand Fixation and Utilization, Liaoning Academy of Agricultural

Sciences, Fuxin 123000, China

[3] Key Laboratory of Regional Environment and Eco-remediation, College of

Environment, Shenyang University, Shenyang 110044, China

[4] University of Chinese Academy of Sciences, Beijing 10049, China

[*] Corresponding author: Tel.: +86 24 83970603; fax: +86 24 83970300.

E-mail address: ruzhenwang@iae.ac.cn (Ruzhen Wang)

**Abstract**

Soil coarseness is the main process decreasing soil organic matter and threatening the productivity of sandy grasslands. Previous studies demonstrated negative effect of soil coarseness on soil carbon storage, but less is known about how soil base cations (exchangeable Ca, Mg, K, and Na) and available micronutrients (available Fe, Mn, Cu, and Zn) response to soil coarseness. In a semi-arid grassland of northern China, a field experiment was initiated in 2011 to mimic the effect of soil coarseness on soil base cations and available micronutrients by mixing soil with different mass proportions of sand: 0% coarse elements (C0), 10% (C10), 30% (C30), 50% (C50), and 70% (C70). Soil coarseness significantly increased soil pH in three soil depths of 0-10 cm, 10-20 cm and 20-40 cm with the highest pH values detected in C50 and C70 treatments. Soil fine particles (smaller than 0.25 mm) significantly decreased with the degree of soil coarseness. Exchangeable Ca and Mg concentrations significantly decreased with soil coarseness degree by up to 29.8% (in C70) and 47.5% (in C70), respectively, across three soil depths. Soil available Fe, Mn and Cu significantly decreased with soil coarseness degree by 62.5%, 45.4% and 44.4%, respectively. As affected by soil coarseness, the increase of soil pH, decrease of soil fine particles (including clay), and decline in soil organic matter were the main driving factors for the decrease of exchangeable base cations (except K) and available micronutrients (except Zn) through soil profile. Developed under soil coarseness, the loss and redistribution of base cations and available micronutrients along soil depths might pose threat to ecosystem productivity of this sandy grassland.

*Key words:* soil degradation, dryland ecosystem, sandy soil, soil texture, Calcium,

Manganese

**1 Introduction**

[revised manuscript text omitted]

County, Liaoning Province, China (Fig. 1). Productive grasslands from Zhanggutai

County have undergone severe desertification due to intense cultivation, overgrazing and increased population (Li et al., 2000; Chen et al., 2005). The soils are susceptible to wind erosion as a result of decrease in plant cover and high annual wind velocity (varies from 3.4 to 4.1 m s$^{-1}$) with frequent occurrence of gales (wind speed > 20 m s$^{-1}$)

(Li et al., 2000). The mean annual temperature is 6.2 °C and mean annual precipitation is about 450 mm which defines the area as semi-arid (Chen et al., 2005).

[revised manuscript text omitted]

Freeze-thaw cycles might promote the leaching of exchangeable Ca, K and Na from surface to subsoil resulting in the unchanged base cations along soil profile. Our results are in contrast with previous studies suggesting that ecosystem were more capable to retain K than other base cations (Nowak et al., 1991; Jobbágy and Jackson,

2001). In this case, it is obvious that many environmental factors, like soil types and plant community composition can be drivers for the vertical distribution of base cations and micronutrients. The leaching of base cations to subsoils might enhance mineral weathering process and pedogenesis by forming kaolinite in topsoils as rapid removing of water-soluble elements (such as exchangeable Ca and Na) (Chadwick and Chorover, 2001). Stronger effect of plant absorption than leaching might contribute to the shallower distribution of exchangeable Mg (Fig. 3b), available Fe (Fig. 4a), Mn (Fig. 4b) and Zn (Fig. 4d). The dominant role of plant cycling in determining the vertical distribution of Mg, Fe, Mn and Zn might illustrate that these elements were scarcer and more limiting nutrients for plant growth in this semi-arid sandy ecosystem (Jobbágy and Jackson, 2001).

**5 Conclusions**

The results showed that grassland soil coarseness decreased soil base cations of exchangeable Ca, Mg and Na as well as available micronutrients of Fe, Mn and Cu.

The loss of SOM, decrease of soil fine particles, and increase of soil pH were the main driving factors for the decrease of base cations and micronutrient availability as affected by soil coarseness. Unchanged concentrations of exchangeable Ca, K and Na along the soil depth might result from the balance between plant cycling and leaching effects. The dominant role of plant cycling over leaching shaped the shallower distribution of exchangeable Mg as well as available Fe, Mn and Zn. The reduction and re-distribution of soil base cations and available micronutrients would potentially influence soil fertility and plant productivity in this desertified grassland ecosystem.

**Author contribution**

Z. Wang, G. Yu, and X. Han designed the experiments; and L. Lü and Y. Zhao carried them out. H. Liu and J. Yin help to do the laboratory analysis. L. Lü and R. Wang prepared the manuscript with contributions from all authors. Y. Jiang helped to revise the manuscript. Mr. Xiao created the figure of our experimental location.

**Acknowledgments**

This work was financially supported by the National Natural Science Foundation of China (41371251).

**References**

Alamusa, Niu C., Zong Q.: Temporal and spatial changes of freeze-thaw cycles in Ulan'aodu Region of Horqin Sandy Land, Northern China in a changing climate, Soil Sci. Soc. Am. J., 78, 89-96, 2014.

[revised manuscript text omitted]

---

## Referee Comment (RC2) · M. van der Ploeg (Referee) · 9 Mar 2016

Review SE-2016-18

The manuscript "Effect of soil coarseness on soil base cations and available micronutrients in a semi-arid sandy grassland" provides an interesting perspective on changes in soil chemistry involved in soil coarseness in China, and under impact of a vegetation transplantation experiment. The manuscript is concise and well-written, and is of interest to Solid Earth. I have some suggestions to improve the impact of the paper.

Main suggestions

1. At present the study is presented a bit technical, I suggest to include a description of the pedogenesis of the area the study was carried out, it makes sense from a

soil origin perspective, because not every soil is prone to effects of soil coarseness. Include proper references to the description of the pedogenesis when these are available (some suggestions below).

2. In the description of the experimental design I have two questions: Page 5 line 13 mentions species composition. What was the composition? And based on the transplantation procedures would the authors expect a difference in start conditions of micronutrients, was it checked? And if there were differences how do the authors expect this to play out over the experimental period of two years?

3. Related to point 2: The vegetation could be a bit more embedded in the discussion with references to other studies in that respect. Where there differences observed in plant composition after two years and how could these have affected the micronutrients?

Specific suggestions

Page 3 line 3 "and are" instead of "as well as"

Page 3 line 14 cause a decrease

Page 3 line 15 decrease of

Page 10 line 17-20 I would suggest to reference Van der Ploeg et al 2012. They show that plant species composition and local variations in chemistry are also related to differences in microtopgraphy. The microtopgraphy may be induced by species differentiation itself. This may be an important factor in the grassland systems here as well, and may also determine differences between the current and other studies in terms of micronutrient species distribution. Another interesting reference in that respect is Burke et al. 1999.

Fig 2 panel c and Fig 3 panel 3: Were no significant differences found, or are the indications for significant difference missing? If first, please mention in caption, if second, please include in Figure.

References

Barthold, F. K., et al. "Land use and climate control the spatial distribution of soil types in the grasslands of Inner Mongolia." Journal of arid environments 88 (2013): 194-205.

Burke, Ingrid C., et al. "Spatial variability of soil properties in the shortgrass steppe: the relative importance of topography, grazing, microsite, and plant species in controlling spatial patterns." Ecosystems 2.5 (1999): 422-438.

Han, Guodong, et al. "Effect of grazing intensity on carbon and nitrogen in soil and vegetation in a meadow steppe in Inner Mongolia." Agriculture, Ecosystems & Environment 125.1 (2008): 21-32.

Sun, Jimin, et al. "Holocene environmental changes in the central Inner Mongolia, based on single-aliquot-quartz optical dating and multi-proxy study of dune sands." Palaeogeography, Palaeoclimatology, Palaeoecology 233.1 (2006): 51-62.

Van der Ploeg, M. J., et al. "Microtopography as a driving mechanism for ecohydrological processes in shallow groundwater systems." Vadose Zone Journal 11.3 (2012).

---

## Author Comment (AC2) · 14 Mar 2016

Comments from Reviewer #1:

Comment: In the Introduction the processes including soil coarsening around the world are briefly mentioned, but they are not mentioned along the text. It would be important to discuss which factors are driving soil coarsening in the study area.

Response: We added information related to soil coarseness in the context. Please see Line 41-42, Line 55-64, Line 78-81, and Line 88-92. And now, the driving factors of soil coarseness in this area can be found in Line 55-59 and Line 90-92.

Comment: In the Study area the authors should induce a reference to some other factors which affect desertification in this region, such as vegetation, lithology (bedrock, sediments?) or aeolian processes (significant?)

Response: The information with reference to other factors has been added. Please find in Line 109-114.

Comment: Are soil layers different along the sections? If it is homogenous, mention it. If soil layer show distinct features, this might have affected your sampling and therefore your experiment. Please clarify it.

Response: We agree with the suggestion. This information has been added in Line 129.

Comment: Another thing which you may have influenced your data at depth are cryogenic processes. The area records freeze-thaw cycles during at least 4-5 months per year. Freeze-thaw cycles affect the vertical structure of the soil through cryoturbation activity. How this process may have affected your data?

Response: The reviewer brings up a good point and we have expanded our speculation related to freeze-thaw cycles. Please see Line 302-305.

Comment: You present nice data about soil base cations and available micronutrients at different depths, but you do not discuss how they influence soil formation processes (pedogenesis). Response: This information has been added in Line 310-313.

Comment: Tables are OK, but the paper would substantially improve with 1/2 new figures including the site location, soil sections, etc.

Response: This figure has been added as Fig. 1. Mr. Xiao helps to create this figure, so we added him as a co-author.

Comment: Line 41 add comma after world; Line 97 change expect to 'expected'; Line 109 after mm, change to 'which defines the area as semi-arid'. Response: These has been corrected in the context. Please see Line 41, Line 100, and Line 115.

Comments from Reviewer #2 (Dr. Martine Van der Ploeg):

[Figure]

Comment: I suggest to include a description of the pedogenesis of the area where the study was carried out.

Response: This kind of description has been added in the context. Please see Line 120-123.

Comment: In the description of the experimental design, I have two questions: Page 5 line 13 mentions species composition. What was the composition? And base on the transplantation species procedures, would the authors expect a difference in start conditions of micronutrients, was it checked? And if there were differences, how do the authors expect this to play out over the experimental period of two years?

Response: Thanks for bringing a nice point. First, the information of species composition was listed in Line 146-149. Second, the conditions of micronutrients would be different among treatments at the beginning because their concentrations were mainly determined by the proportion of clay or SOM concentration. And species composition was the same for all treatment plots at the beginning which would not influence micronutrients at this time (this information has been added in Line 146). The role of soil fine particles and SOM concentration in shaping micronutrient distribution has been discussed in Line 261-287. Frankly, we did not check the conditions of micronutrients at the start of the experiment. In our further work, we will analyze micronutrients in both plants and soils to see the response of soil micronutrients to plant uptake along with the experiment going on.

Comment: Related to the previous comment: The vegetation could be a bit more embedded in the discussion with reference to other studies in that aspect. Were there differences observed in plant composition after two years and how could these have affected the micronutrients?

Response: Thanks for this comment. Actually, we think net primary production can have greater impact on availabilities of soil base cations and micronutrients than plant community composition under different soil coarseness degree. Because higher plant biomass and higher plant nutrient demands would induce mobilization of these nutrients. Related information has been added in Line 281-284.

Comment: Page 3 line 3 "and are" instead of "as well as"

Response: This has been corrected. Please see Line 67.

Comment: Page 3 line 14 cause a decrease

Response: This has been corrected. See Line 78.

Comment: Page 3 line 15 decrease of

Response: This has been corrected. Please see Line 79.

Comment: Page 10 line 17-20 I would suggest to reference Van der Ploeg et al. 2012. They show that plant species composition and local variations in chemistry are also related to differences in microtopgraphy. The microtopgraphy may be induced by species differentiation itself. This may be an important factor in the grassland systems here as well, and may also determine difference between the current and other studies in terms of micronutrient species distribution. Another interesting reference in that respect is Burke et al. 1999.

Response: Nice point. We have also cited these two references here. Please see Line 310.

Comment: Fig. 2 panel c and Fig. 3 panel d: Were there no differences found, or are the indications for significant difference missing? If first, please mention in caption, if second, please include in Figure.

Response: There were no differences found. We have mentioned this in the figure captions. Please see Line 517, 525.

Please also note the supplement to this comment:
http://www.solid-earth-discuss.net/se-2016-18/se-2016-18-AC2-supplement.pdf

N

▲ **Zhanggutai County**

**Liaoning Province**

50 100    200
█████▌ ░ ██████ KM

**Fig. 1.** Location of the experimental site.

[Figure]

**Fig. 2.** Soil pH values for three soil depths (a) and proportion of soil fine particles (< 0.25 mm) for 0-10 cm soil in different soil coarseness degrees of 0% sand addition (C0), 10% (C10), 30% (C30), 50% (C5

[Figure]

Fig. 3. Soil base cations of exchangeable Ca (a), Mg (b), K (c) and Na (d) for three soil depths in different soil coarseness degrees of 0% sand addition (C0), 10% (C10), 30% (C30), 50% (C50) and 70% (C70). D

[Figure]

**Fig. 4.** Soil available micronutrients of available Fe(a), Mn (b), Cu (c) and Zn (d) for three soil depths in different soil coarseness degrees of 0% sand addition (C0), 10% (C10), 30% (C30), 50% (C50) and 70%

[Figure]

**Fig. 5.** Fig. S1 Proportion of soil clay particles for 0-10 cm soil in different soil coarseness degrees of 0% sand addition (C0), 10% (C10), 30% (C30), 50% (C50) and 70% (C70). Data represent mean ± SE (n=6).

**Supplement:**

**Effect of soil coarseness on soil base cations and available**

**micronutrients in a semi-arid sandy grassland**

Linyou Lü[1,2], Ruzhen Wang[1,*], Heyong Liu[1,3], Jinfei Yin[1,4], Jiangtao Xiao[1,4],

Zhengwen Wang[1], Yan Zhao[2], Guoqing Yu[2], Xingguo Han[1], Yong Jiang[1]

[1] State Key Laboratory of Forest and Soil Ecology, Institute of Applied Ecology,

Chinese Academy of Sciences, Shenyang 110016, China

[2] Institute of Sandyland Improvement and Utilization, Liaoning Academy of

Agricultural Sciences, Fuxin 123000, China

[3] Key Laboratory of Regional Environment and Eco-remediation, College of

Environment, Shenyang University, Shenyang 110044, China

[4] University of Chinese Academy of Sciences, Beijing 10049, China

[*] Corresponding author: Tel.: +86 24 83970603; fax: +86 24 83970300.

 E-mail address: ruzhenwang@iae.ac.cn (Ruzhen Wang)

**Abstract**

Soil coarseness is the main process decreasing soil organic matter and threatening the productivity of sandy grasslands. Previous studies demonstrated negative effect of soil coarseness on soil carbon storage, but less is known about how soil base cations (exchangeable Ca, Mg, K, and Na) and available micronutrients (available Fe, Mn, Cu, and Zn) response to soil coarseness. In a semi-arid grassland of northern China, a field experiment was initiated in 2011 to mimic the effect of soil coarseness on soil base cations and available micronutrients by mixing soil with different mass proportions of sand: 0% coarse elements (C0), 10% (C10), 30% (C30), 50% (C50), and 70% (C70). Soil coarseness significantly increased soil pH in three soil depths of 0-10 cm, 10-20 cm and 20-40 cm with the highest pH values detected in C50 and C70 treatments. Soil fine particles (smaller than 0.25 mm) significantly decreased with the degree of soil coarseness. Exchangeable Ca and Mg concentrations significantly decreased with soil coarseness degree by up to 29.8% (in C70) and 47.5% (in C70), respectively, across three soil depths. Soil available Fe, Mn and Cu significantly decreased with soil coarseness degree by 62.5%, 45.4% and 44.4%, respectively. As affected by soil coarseness, the increase of soil pH, decrease of soil fine particles (including clay), and decline in soil organic matter were the main driving factors for the decrease of exchangeable base cations (except K) and available micronutrients (except Zn) through soil profile. Developed under soil coarseness, the loss and redistribution of base cations and available micronutrients along soil depths might pose threat to ecosystem productivity of this sandy grassland.

*Key words:* soil degradation, dryland ecosystem, sandy soil, soil texture, Calcium,

Manganese

**1 Introduction**

[revised manuscript text omitted]

County, Liaoning Province, China (Fig. 1). Productive grasslands from Zhanggutai

County have undergone severe desertification due to intense cultivation, overgrazing and increased population (Li et al., 2000; Chen et al., 2005). The soils are susceptible to wind erosion as a result of decrease in plant cover and high annual wind velocity (varies from 3.4 to 4.1 m s$^{-1}$) with frequent occurrence of gales (wind speed > 20 m s$^{-1}$)

(Li et al., 2000). The mean annual temperature is 6.2 °C and mean annual precipitation is about 450 mm which defines the area as semi-arid

 (Chen et al., 2005). The frost-free period lasts approximately 150 days (Chen et al., 2005). Soil texture of the experiment site is sandy soil with 99.32 ±0.13 % sand,

0.45 ±0.14 % silt, and 0.23 ±0.02 % clay (means ±standard deviation, data measured from control soil). The soil type is classified as a *Aeolic Eutric Arenosol*

according to the FAO classification (IUSS Working Group WRB, 2014). *Arenosols*

*are mainly developed in sand dune areas which are featured by a sandy texture and*

*low soil organic carbon (SOC) concentrations and prone to soil coarsening* (Barthold et al., 2013). This area constitutes an agro-pastoral ecotone which is severely degraded due to excessive cultivation and grazing (Chen et al., 2005).

**2.2 Experimental design**

In May 2011, a complete randomized design was applied to the site. Within a 24 m ×

29 m area (696 m$^2$), thirty 4 m×4 m plots were established for five treatments with six replicates per treatment. The soil within this experimental area is homogenous.

[revised manuscript text omitted]